# Criticality and degeneracy in injury-induced changes in primary afferent excitability and the implications for neuropathic pain

Stéphanie Ratté[1,2,3], Yi Zhu[3], Kwan Yeop Lee[1], Steven A Prescott[1,2,3]*

[1]Neurosciences and Mental Health, The Hospital for Sick Children, Toronto, Canada; [2]Department of Physiology, University of Toronto, Toronto, Canada; [3]Pittsburgh Center for Pain Research, University of Pittsburgh, Pittsburgh, United States

**Abstract** Neuropathic pain remains notoriously difficult to treat despite numerous drug targets. Here, we offer a novel explanation for this intractability. Computer simulations predicted that qualitative changes in primary afferent excitability linked to neuropathic pain arise through a switch in spike initiation dynamics when molecular pathologies reach a tipping point (criticality), and that this tipping point can be reached via several different molecular pathologies (degeneracy). We experimentally tested these predictions by pharmacologically blocking native conductances and/or electrophysiologically inserting virtual conductances. Multiple different manipulations successfully reproduced or reversed neuropathic changes in primary afferents from naïve or nerve-injured rats, respectively, thus confirming the predicted criticality and its degenerate basis. Degeneracy means that several different molecular pathologies are individually sufficient to cause hyperexcitability, and because several such pathologies co-occur after nerve injury, that no single pathology is uniquely necessary. Consequently, single-target-drugs can be circumvented by maladaptive plasticity in any one of several ion channels.

*For correspondence: steve.
prescott@sickkids.ca

**Competing interests:** The authors declare that no competing interests exist.

**Reviewing editor**: Ronald L Calabrese, Emory University, United States

## Introduction

Neuropathic pain—pain arising from damage to or dysfunction of the nervous system (*Merskey and Bogduk, 1994*)—is notoriously difficult to treat. Effective treatments have eluded discovery despite intense research and many promising leads (*Woolf, 2010*). Proposed explanations for the lack of clinical translation have focused on preclinical animal models (*Rice et al., 2008*; *Mogil, 2009*) and on clinical trial design (*Dworkin et al., 2011*; *Mao, 2012*). An alternative possibility is that degeneracy within the pain system allows the pathogenic process to circumvent single-target-drugs. If true, the single-target-drug paradigm is bound to fail no matter how good the animal models or clinical trials are, and a new paradigm is needed.

Degeneracy refers to multiple 'different' mechanisms conveying equivalent function (*Edelman and Gally, 2001*); by comparison, redundancy refers to multiple instantiations of the 'same' mechanism. Both convey robustness to complex systems (*Kitano, 2004a*). But if a pathogenic process were to hijack degeneracy, the pathological state could itself become robust, or in other words resistant to treatment. Accordingly, degeneracy is recognized as an important factor for cancer and other complex diseases (*Kitano, 2004a, 2004b*; *Tian et al., 2011*), including epilepsy (*Klassen et al., 2011*) but has yet to inform pain research or analgesic drug development. That said, degeneracy has been recognized in neural systems (*Prinz et al., 2004*) and its existence and functional implications are gaining increasing attention (*Grashow et al., 2009, 2010*; *Marder, 2011*; *Amendola et al., 2012*; *Zhao and Golowasch, 2012*; *Gutierrez et al., 2013*; *O'Leary et al., 2013*; *Ransdell et al., 2013*).

**eLife digest** Although the pain associated with an injury is unpleasant, it normally serves an important purpose: to make you avoid its source. However, some pain appears to arise from nowhere. Frustratingly, this type of pain, known as neuropathic pain, does not respond to common painkillers and is thus very difficult to treat.

The neurons that transmit pain and other sensory information do so using electrical signals. In response to a stimulus, ions travel through channels in the membrane of a neuron, which leads to a change in the electrical potential of the membrane. When this change is large enough, a voltage spike is produced: this signal is ultimately transmitted to the brain.

When certain neurons fire too easily or too often, neuropathic pain can arise. This hyperexcitability can make something painful feel even worse, or it can make things hurt that shouldn't. To prevent this, extensive research has been devoted to identify drugs that target particular types of ion channels and block them. However, despite the discovery of many promising drugs, those drugs have been frustratingly ineffective in clinical trials.

Using simulations and experiments, Ratté et al. have examined the behavior of a type of neuron that normally conducts information about touch, but the brain sometimes misinterprets this information as pain. Increasing the flow of ions through the cell membrane in these simulations eventually causes a 'tipping point' to be crossed, which triggers a dramatic, discontinuous change in spiking pattern. However, as several different types of ion channels contribute to the current, there are several different ways in which the tipping point can be crossed.

This ability to produce the same result by multiple means is a common feature of complex systems. Known as degeneracy, it makes systems more robust, as a given result can still be achieved if one particular attempt to achieve this result fails. The work of Ratté et al. helps to explain why drugs that target just one type of ion channel may fail to relieve neuropathic pain: maladaptive changes in any one of several other ion channels may circumvent the therapeutic effect.

A degenerate basis for neuropathic pain is suggested by basic research findings. For example, increased function of the sodium channel $Na_v1.8$ is sufficient to produce the hypersensitivity associated with neuropathic pain (*Bierhaus et al., 2012*; *Wu et al., 2012*). $Na_v1.8$-targetted manipulations predictably fail in $Na_v1.8$ knock-out animals, but those animals can nonetheless develop injury-induced hypersensitivity (*Nassar et al., 2005*). This indicates that injury-induced changes in other ion channels— and indeed many such changes occur (for reviews, see *Campbell and Meyer, 2006*; *Woolf and Ma, 2007*; *Basbaum et al., 2009*; *Marchand et al., 2009*; *Gold and Gebhart, 2010*)—are also sufficient to cause neuropathic pain.

To directly explore degeneracy in the context of neuropathic pain, we tested whether multiple distinct molecular pathologies are sufficient to produce the cellular hyperexcitability associated with neuropathic pain. Hyperexcitability characterized by quantitative changes such as reduced threshold and qualitative changes such as altered spiking patterns (see below) develops at multiple points along the neuraxis (*Costigan et al., 2009*). This includes primary somatosensory afferents whose spontaneous spiking and exaggerated responsiveness are thought to underlie spontaneous pain and hypersensitivity, respectively (for review, see *Devor, 2005*). Notably, peripheral input helps drive central sensitization (*Devor, 1991*; *Gracely et al., 1992*; *Koltzenburg et al., 1994*), meaning increased peripheral input is amplified rather than attenuated centrally.

Injury-induced hyperexcitability is not limited to nociceptors; on the contrary, hyperexcitability also develops in myelinated afferents that normally convey innocuous information but that contribute to mechanical allodynia (hypersensitivity) under neuropathic conditions (*Campbell et al., 1988*; *Koltzenburg et al., 1994*; *Devor, 2009*; *King et al., 2011*). Hyperexcitability in myelinated afferents is characterized by a triad of qualitative changes that include repetitive spiking, membrane potential oscillations (MPOs), and bursting (*Liu et al., 2000*; *Herzog et al., 2001*; *Xing et al., 2001*; *Liu et al., 2002*; *Ma and LaMotte, 2007*; *Fan et al., 2011*; *Xie et al., 2011*; *Song et al., 2012*). We refer to this qualitatively altered excitability as 'neuropathic'. We recently showed through mathematical modeling that all three neuropathic changes arise from a switch in spike initiation dynamics (*Rho and Prescott, 2012*); there is, therefore, a one-to-many mapping between altered spike initiation dynamics and

qualitative changes in cellular excitability. On the other hand, we found a many-to-one mapping between parameter values (whose variations represent injury-induced molecular changes) and cellular excitability; moreover, continuous parameter variations led to a discontinuous change in spike initiation dynamics. The many-to-one mapping constitutes degeneracy and the discontinuity, or tipping point, reflects criticality. The results identify spike initiation as a key nonlinear process whose qualitative alteration explains multiple features of cellular hyperexcitability on the basis of several possible molecular pathologies.

In the current study, we experimentally tested our theory. After identifying the activation properties of currents affecting spike initiation, we used nonlinear dynamical analysis of our mathematical model to define the system's tipping point. From this, we generated several predictions as to how that tipping point could be crossed. To experimentally test those predictions, we applied different manipulations designed to force neurons in one or the other direction across their tipping point, therein acutely reproducing neuropathic excitability in primary afferent neurons from naive animals or acutely reversing neuropathic excitability in neurons from nerve-injured animals. Manipulations involved decreasing and/or increasing subthreshold currents via pharmacology and/or dynamic clamp, respectively, in identified myelinated primary afferents. All predictions were confirmed, therein supporting our theory that primary afferent hyperexcitability arises through a critical transition whose molecular basis is highly degenerate.

## Results

### Theory and modeling

According to our previous theoretical work, spike initiation occurs through a time- and voltage-dependent competition between *net* fast-activating inward current and *net* slower-activating outward current (*Prescott et al., 2008*). Because multiple types of ion channels contribute to each *net* current—currents with similar kinetics sum linearly (*Kepler et al., 1992*)—injury-induced changes in any one of the contributing ion channels can bias the competition (*Rho and Prescott, 2012*). Using a Morris–Lecar model that comprises only two variables whose interaction is sufficient to produce spikes, we adjusted parameters ('Materials and methods') to give a spike threshold approximating that observed in the soma of myelinated afferents, which is about −35 mV. The parameters of this base model were thereafter unchanged. Next, we derived the voltage-dependency and kinetics (*Figure 1A*) for an additional conductance (*Equations 1–4* in 'Materials and methods') which, when added to our base model, modulates its spike initiation dynamics. This conductance corresponds to either a sodium or potassium channel depending on its associated reversal potential and, according to our analysis, must be active at subthreshold voltages. To be clear, the parameters for this conductance were chosen to modulate spike initiation dynamics, not to model specific ion channel types that are known to be altered by nerve injury. That said, although our determination of parameters was agnostic to molecular identities, parameter values resemble those of known channel types, as noted in subsequent sections. We then varied the maximal conductance of the additional conductance(s). Based on two-parameter bifurcation analysis, we determined the combinations of sodium conductance $\bar{g}_{Na}$ and potassium conductance $\bar{g}_K$ required to give repetitive spiking at different stimulus thresholds (*Figure 1B*). Somata of myelinated primary afferents normally generate a single spike at stimulus onset, which corresponds to the gray-shaded region, whereas a subset of those neurons spike repetitively after nerve injury (e.g., *Liu et al., 2002*), which corresponds to the colored region.

Neuropathic changes in excitability arise from a switch in spike initiation dynamics (*Rho and Prescott, 2012*; *Figure 1—figure supplement 1*). In brief, the colored region of *Figure 1B* corresponds to a parameter regime in which a subcritical Hopf bifurcation occurs when stimulus intensity $I_{stim}$ reaches a critical value $I^*$. A Hopf bifurcation represents destabilization of the 'resting' state: repetitive spiking occurs when $I_{stim}$ exceeds $I^*$, membrane potential oscillations (MPOs) arise in the presence of noise when $I_{stim}$ approaches $I^*$, and bursting occurs when adaptation $I_{adapt}$ causes $I_{stim}-I_{adapt}$ to sweep back and forth across $I^*$. Outside the colored region of *Figure 1B*, spike initiation is effectively limited to a quasi-separatrix crossing; according to this mechanism, the resting state remains stable but a single spike is produced if the system transiently escapes from that state during a stimulus transient. In the absence of a Hopf bifurcation, repetitive spiking, MPOs and bursting are simply not possible.

The boundary between gray and blue regions in our colored 'excitability' diagram (*Figure 1B*), thus represents the critical tipping point separating normal and neuropathic parameter regimes. That tipping point is represented by a simple curve in subsequent excitability diagrams. We predict that a cell with

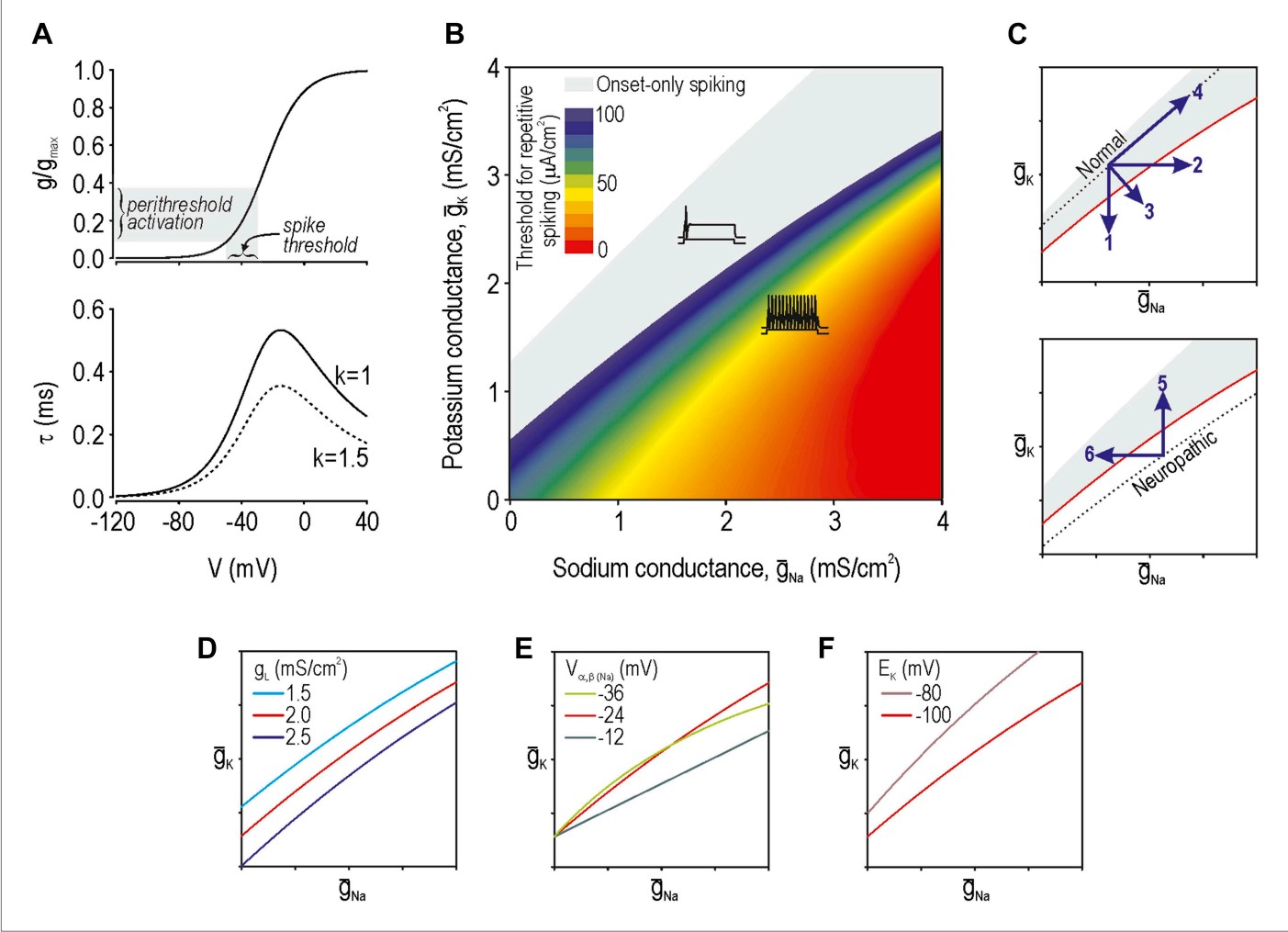

**Figure 1.** Modeling and theory. (**A**) Voltage-dependency and kinetics of subthreshold conductance. In bottom panel, $\tau = (\alpha+\beta)^{-1}$ where $\alpha$ and $\beta$ are defined in *Equations 3* and *4*. Equivalent activation parameters were used to implement either a sodium or potassium current based on reversal potential. (**B**) Two-parameter bifurcation analysis in which $\bar{g}_{Na}$ and $\bar{g}_{K}$ were co-varied to determine the $\bar{g}_{Na}:\bar{g}_{K}$ ratio for a given threshold for repetitive spiking. Gray shading shows parameter regime associated with onset-only spiking. Changes in excitability are explained by a switch in spike initiation dynamics (*Figure 1—figure supplement 1*). (**C**) Predictions (indicated by numbered arrows) of how manipulating $\bar{g}_{Na}$ and/or $\bar{g}_{K}$ may force the system across a tipping point, thus reproducing or reversing neuropathic excitability. Position of the tipping point depends on many other parameters including leak conductance (**D**), voltage-dependency of $\bar{g}_{Na}$ and $\bar{g}_{K}$ (**E**), and potassium reversal potential (**F**).

The following figure supplements are available for figure 1:

**Figure supplement 1**. Spike initiation dynamics differ between normal and neuropathic conditions.

normal excitability can be converted to neuropathic excitability (i.e., forced across its tipping point) by a decrease in $\bar{g}_{K}$, an increase in $\bar{g}_{Na}$, or some combination thereof (arrows 1–3 on *Figure 1C*). However, balanced conductance changes may offset one another, resulting in no qualitative alteration of excitability (arrow 4). Contrariwise, the inverse changes are predicted to normalize excitability in a hyperexcitable cell by forcing the system in the opposite direction across its tipping point (arrows 5 and 6). Each arrow corresponds to a prediction tested in the experimental portion of this study.

Changes in other conductances, in parameters other than maximal conductance (e.g., activation properties), or in parameters not strictly connected to conductance (e.g., reversal potential) can all affect where the tipping point lies (*Figure 1D–F*, respectively). It is not feasible to experimentally test all parameter variations and combinations thereof, and hence we focused on predictions outlined in *Figure 1C*. But one should bear in mind that the tipping point in a real neuron will depend on numerous

parameters and thus correspond to a boundary within a high-dimensional space. The important considerations are (1) that that boundary exists (which implies criticality), and (2) that there are many ways to cross it (which implies degeneracy). In other words, our theory predicts that excitability can change abruptly on the basis of many different molecular changes when and if those molecular changes reach a tipping point.

## Reproducing neuropathic excitability in neurons from naive animals

Our next step was to test our predictions experimentally. Unlike typical experiments in which excitability is compared between different neurons before and after nerve injury, and where it is impossible to account for all injury-induced molecular changes and between-cell differences, we compared excitability in the same neuron before and after strictly controlled manipulations dictated by our simulation results. Furthermore, rather than trying to reproduce a complete set of injury-induced molecular pathologies, our manipulations were designed to reproduce one or two of those pathologies at a time. This approach reveals which molecular pathologies, alone or together, are sufficient to reproduce neuropathic excitability without unaccounted for co-variations in other parameters. Furthermore, by applying manipulations acutely, our approach avoids the confounding influence of compensatory changes that might develop over longer time scales.

We targeted myelinated afferents (somatic diameter ≥30 µm) because they are directly implicated in allodynia and because they have been reported to develop the triad of neuropathic excitability changes described in the 'Introduction'. We initially targeted myelinated cutaneous afferents retrogradely labeled by DiI injected intradermally into the hindpaw ('Materials and methods') because this population comprises low-threshold mechanosensors that may subserve mechanical allodynia; we subsequently tested labeled muscle afferents, as reported at the end of the 'Results' section. In keeping with previous studies, neurons harvested from naive, uninjured rats responded to a square pulse of current injection with onset-only spiking, even at high stimulus intensities (*Figure 2A*; n = 23 cutaneous afferents). Neurons could spike again in response to subsequent increments in stimulus intensity (*Figure 2A*, inset), thus ruling out complete sodium channel inactivation as the basis for non-repetitive spiking. This spiking pattern places naive neurons within the gray-shaded region of our excitability diagram.

Our first prediction was that decreasing subthreshold potassium conductance would reproduce neuropathic changes in excitability, consistent with injury-induced reduction of $K_v1$ channels (*Everill and Kocsis, 1999*; *Ishikawa et al., 1999*; *Kim et al., 2002*; *Hammer et al., 2010*; *Zhao et al., 2013*). To test this, we applied 4-aminopyridine (4-AP) with a maximal concentration between 1.5 mM and 5 mM to decrease the total potassium current activated at subthreshold voltages. As drug concentration increased during wash-in, MPOs developed followed by a switch to repetitive spiking during stimulation; the reverse sequence occurred during washout (*Figure 2B*). The broad peak in the power spectrum (*Figure 2B*, inset) shows that pharmacologically induced MPOs are consistent with previous descriptions of injury-induced MPOs (*Song et al., 2012*) and with a noise-dependent mechanism (*Rho and Prescott, 2012*). In 6 of 11 neurons tested, MPOs and repetitive spiking co-developed during wash-in of 4-AP, consistent with their common connection with the subcritical Hopf bifurcation. Bursting was not observed (see below). Using 5 nM α-dendrotoxin, which more selectively blocks $K_v1$ channels, we obtained equivalent results in 2 of 4 additional neurons (data not shown). Reproduction of neuropathic excitability via potassium channel blockade in a total of 8 of 15 neurons represents a significantly higher conversion rate than expected by chance (i.e., 0 of 23 neurons, as determined from the excitability at the start and end of recordings in each neuron; p<0.001; Fisher's exact test).

Our second prediction was that increasing subthreshold sodium conductance would reproduce neuropathic excitability, consistent with injury-induced increase of $Na_v1.3$ (*Waxman et al., 1994*; *Kim et al., 2001*; *Fukuoka et al., 2008*; *Huang et al., 2008*). To test this, we added a virtual sodium conductance with the activation properties described in *Figure 1A*. As expected, increasing the virtual sodium current produced MPOs and repetitive spiking (*Figure 2C*). Power spectral analysis (*Figure 2C*, inset) shows that dynamic clamp-induced MPOs are comparable to injury- and pharmacologically-induced MPOs. In 9 of 19 neurons tested, MPOs and repetitive spiking co-developed when sufficient virtual sodium conductance was inserted; this conversion rate is significantly higher than expected by chance (p<0.001; Fisher's exact test). Of the 10 neurons that did not develop repetitive spiking, five nonetheless developed MPOs. Again, bursting was not observed.

Before proceeding to test predictions 3 and 4, we sought to explain the absence of bursting. Our modeling indicated that although the subcritical Hopf bifurcation is necessary to create a stimulus

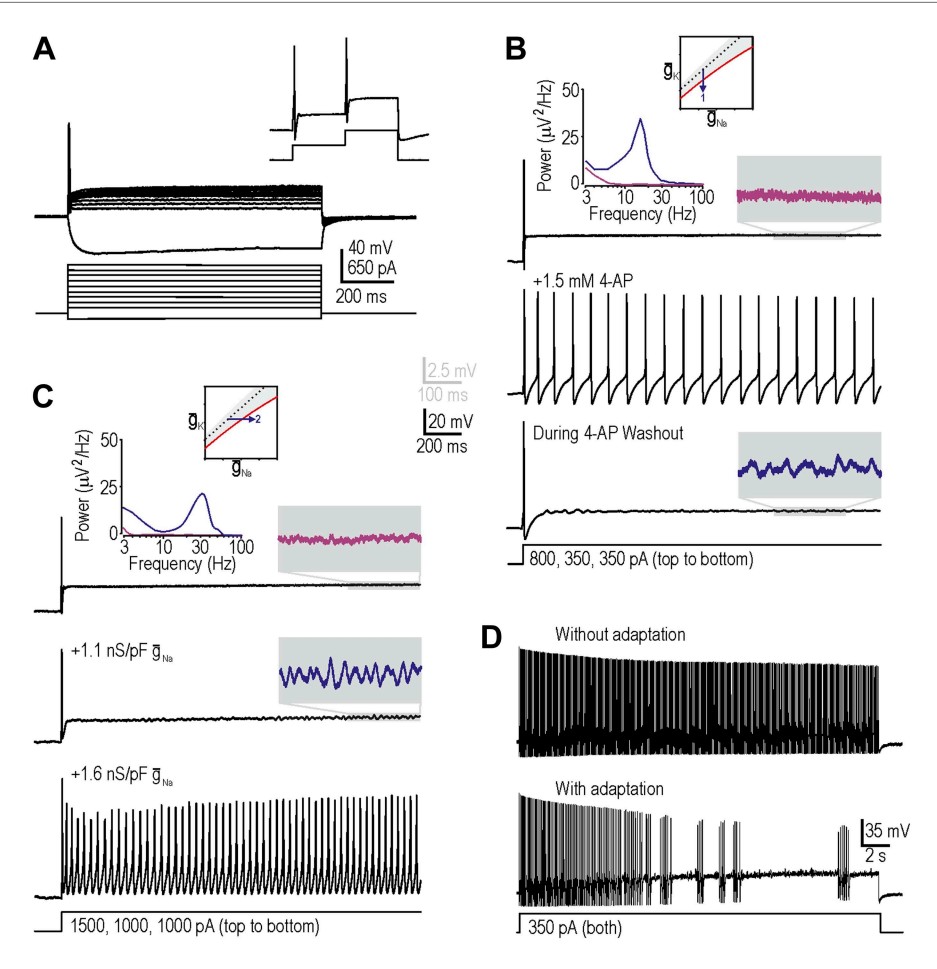

**Figure 2**. Reproduction of neuropathic excitability in naive cutaneous neurons. (**A**) Medium- to large-diameter (putative myelinated) cutaneous afferents from naive rats respond to depolarization with a single spike at stimulus onset. Neurons can respond to increments in stimulus intensity (inset), which demonstrates that sodium channels are not completely inactivated after the initial spike. (**B**) Prediction 1 was tested by blocking potassium current by application of 1.5–5 mM 4-AP. By repeating stimulation during wash-in and/or wash-out of the drug, we observed repetitive spiking and MPOs (in the order expected based on the change in drug concentration) in response to equivalent stimulation. Gray-shaded windows show enlarged views of voltage traces, with colors corresponding to the power spectra shown at the top. (**C**) Prediction 2 was tested by adding virtual sodium conductance via dynamic clamp. As predicted, MPOs and repetitive spiking developed as virtual sodium conductance was increased, but bursting was absent. Note that a conductance density of 1 nS/pF corresponds to 1 mS/cm² assuming a specific membrane capacitance of 1 µF/cm², which means that the conductance densities added by dynamic clamp are the same order of magnitude as those predicted by modeling in **Figure 1B**. (**D**) Bursting was achieved in repetitively spiking neurons by introducing an AHP-type adaptation current via dynamic clamp.

range in which the system is bistable (i.e., in which quiescence or repetitive spiking are both possible), bursting also requires a slow process like spike-dependent adaptation to sweep the system back and forth across the bistable region, thus allowing hysteresis to manifest bursting; in other words, bursting requires a subcritical Hopf bifurcation and adaptation (**Rho and Prescott, 2012**). We therefore hypothesized that our repetitively spiking neurons, although capable of bursting, lacked the required adaptation. To test this, we inserted a virtual adaptation current ('Materials and methods'). As expected, this enabled bursting under conditions associated with repetitive spiking in all three cells tested (**Figure 2D**) but had no effect in the same three cells under normal conditions with onset-only spiking (data not shown). These data confirm that bursting would have co-developed with repetitive spiking and MPOs if adaptation currents had been present, and suggest that adaptation currents are upregulated after nerve injury

(possibly as a compensatory measure) rather than existing occultly in naive neurons whose spiking is already transient.

## Additivity and subtractivity of conductance changes

In a subset of neurons, neither decreasing $K_v1$ potassium conductance nor increasing $Na_v1.3$-like sodium conductance reproduced neuropathic excitability. Larger manipulations may have succeeded in forcing neurons across their tipping point but technical considerations (e.g., off-target effects at high drug concentrations and recording instability for large virtual conductances) limited the magnitude of manipulations. But as explained in *Figure 1*, variation of a single conductance is not the only way to force a neuron across its tipping point; instead, small changes in more than one conductance may combine to produce a net conductance change that is large enough to force the neuron across its tipping point. As an aside, if multiple small changes can recombine in different ways (which is plausible under conditions in which dozens of different ion channels are up- or down-regulated), then degeneracy could exist on the basis of many different combinations being sufficient to produce neuropathic excitability; in other words, no one set of combined changes would be uniquely necessary just as no single change is uniquely necessary. To explore how conductance changes combine, we proceeded with testing predictions 3 and 4.

Our third prediction was that manipulations tested independently in predictions 1 and 2 could combine to reproduce neuropathic changes in excitability; indeed, in 8 neurons in which neither decreasing potassium conductance nor increasing sodium conductance was sufficient to produce repetitive spiking and MPOs, we tested the two manipulations together and found that the combination was sufficient to produce hyperexcitability in 7 of them (*Figure 3A*). We also observed that the minimum virtual sodium conductance needed to reproduce neuropathic excitability was reduced during 4-AP application (*Figure 3B*). We quantified the degree to which potassium channel blockade had moved the system towards its tipping point by comparing the minimum virtual sodium conductance needed to force the system across its tipping point without vs with 4-AP. The median (and 25–75 percentile range) virtual sodium conductance was significantly reduced from 0.51 (range 0.47–0.63) nS/pF without 4-AP to 0.23 (range 0.13–0.40) nS/pF with 4-AP (p<0.001; Mann–Whitney U test) (*Figure 3C*). Non-parametric statistics were used because of the non-Gaussian distribution of data points.

Our fourth prediction was that different manipulations could offset one another, resulting in no net change in excitability. To test this, we first tested whether inserting a virtual potassium conductance could reverse the hyperexcitability induced by blockade of native potassium conductance by 4-AP. Results confirmed our prediction in 3 of 3 neurons tested (*Figure 3D*), thus demonstrating the specificity of the 4-AP effect; in other words, even if channels other than $K_v1$ were blocked by 4-AP, the effect of 4-AP on excitability is attributable to blockade of subthreshold potassium current given that reintroducing that type of current reverses the altered excitability. More interesting is the observation that hyperexcitability caused by insertion of a virtual sodium conductance was reversed by insertion of a virtual potassium conductance in 3 of 3 neurons tested (*Figure 3E*). These data confirm prediction 4 and demonstrate the subtractivity of different manipulations.

## Reversing neuropathic excitability in neurons from nerve-injured animals

In our final two predictions, we sought to move neurons in the opposite direction across their tipping point, which required neurons rendered hyperexcitable by nerve injury ('Materials and methods'). Surprisingly, only 1 of 9 injured cutaneous afferents exhibited any degree of repetitive spiking defined here as at least three spikes during sustained stimulation, which is not a significant change compared to chance (p=0.28; Fisher's exact test compared to spontaneous conversion rate of 0 in 23 cutaneous afferents). By comparison, 6 of 9 injured neurons not labeled by intradermal injection of DiI exhibited neuropathic excitability, an example of which is shown in *Figure 4A* (p<0.001; Fisher's exact test compared to 1 in 9 cutaneous). However, because the identity of unlabeled neurons is uncertain and given that previous studies showing that muscle afferents are in fact more prone than cutaneous afferents to becoming grossly hyperexcitable (*Michaelis et al., 2000*; *Liu et al., 2002*), we retrogradely labeled muscle afferents. 11 of 20 injured muscle afferents exhibited neuropathic excitability (*Figure 4B*), which is a significantly higher proportion than 1 in 9 injured cutaneous afferents (p<0.05; Fisher's exact test) and the spontaneous conversion rate of 0 in 27 muscle afferents from naive animals (p<0.001; Fisher's exact test). Neuropathic excitability was reversed by insertion of virtual potassium conductance in 10 of 10 neurons tested or by reduction of sodium conductance via application of 10 µM riluzole in 4 of 4 neurons tested, thus confirming predictions 5 and 6, respectively (*Figure 4A,B*). In the injured muscle

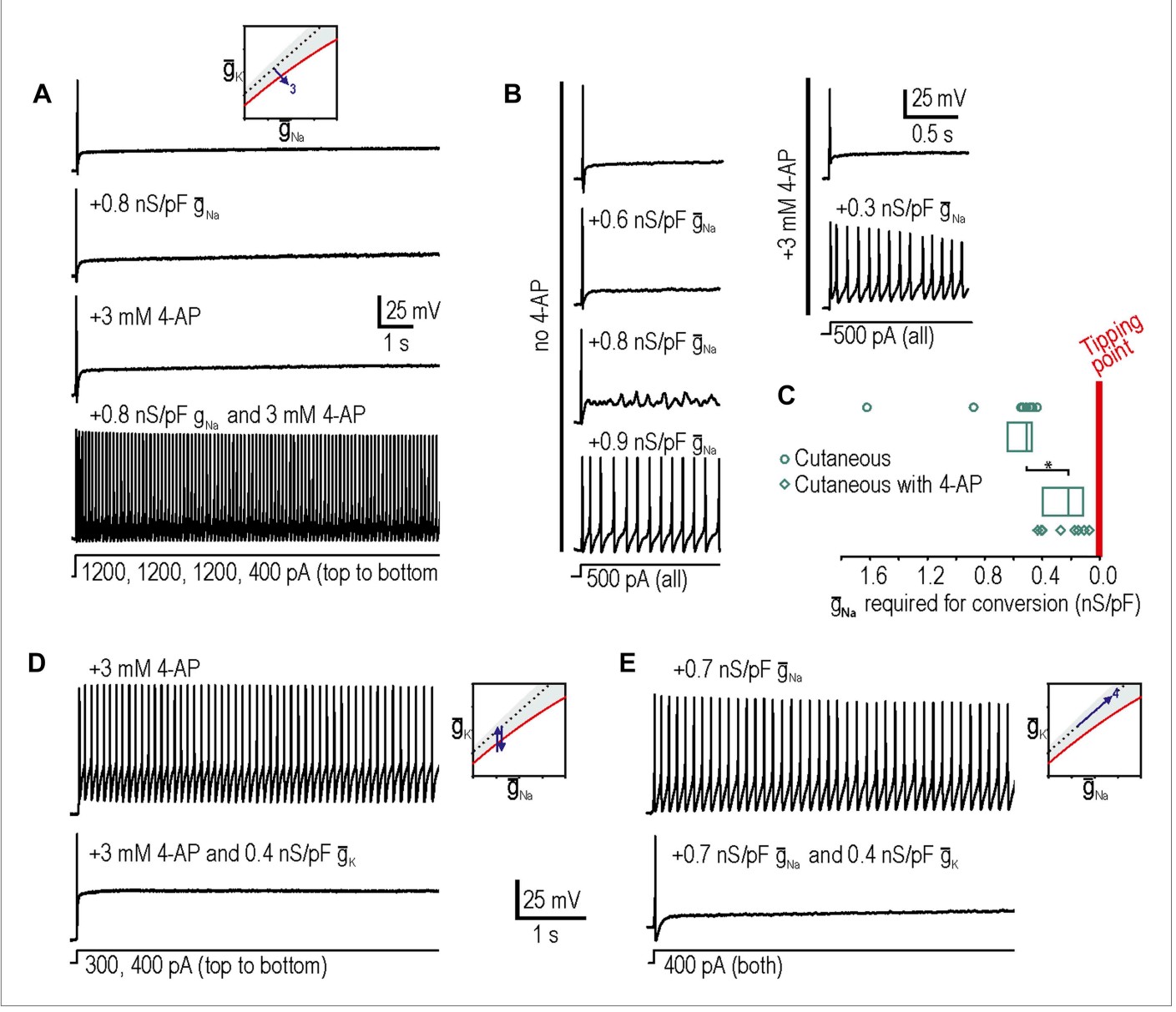

**Figure 3**. Additivity and subtractivity of conductance changes. (**A**) Example in which neither virtual sodium conductance nor potassium channel blockade reproduced neuropathic excitability, whereas the combination of manipulations did, thus confirming prediction 3, that manipulations can be additive. (**B**) Example in which the minimum virtual sodium conductance required to reproduce neuropathic excitability was reduced when combined with potassium channel blockade. (**C**) Even when 4-AP application did not, on its own, produce repetitive spiking, it nonetheless moved the neuron significantly closer to criticality as evidenced by comparing the minimal virtual sodium conductance needed to produce repetitive spiking without vs with 4-AP (*p<0.001; Mann–Whitney U test). Points represent data from individual neurons and boxes represent median and 25–75 percentile range. Repetitive spiking caused by 4-AP application (**D**) or by virtual sodium conductance (**E**) was reversed by insertion of virtual potassium conductance. Panel **E** confirms prediction 4, that manipulations can be subtractive.

afferents whose neuropathic excitability was reversed by insertion of virtual potassium conductance, the median conductance (and 25–75 percentile range) needed to cause reversal was only 0.14 (range 0.12–0.15) nS/pF. This suggests that the neuropathic state lies close to the tipping point, which is consistent with our 100% success rate in reversing neuropathic excitability by manipulating either potassium or sodium conductance (see above).

The concept of measuring proximity to the tipping point prompted us to ask whether cutaneous afferents, only one of which exhibited grossly abnormal excitability after nerve injury, were nonetheless

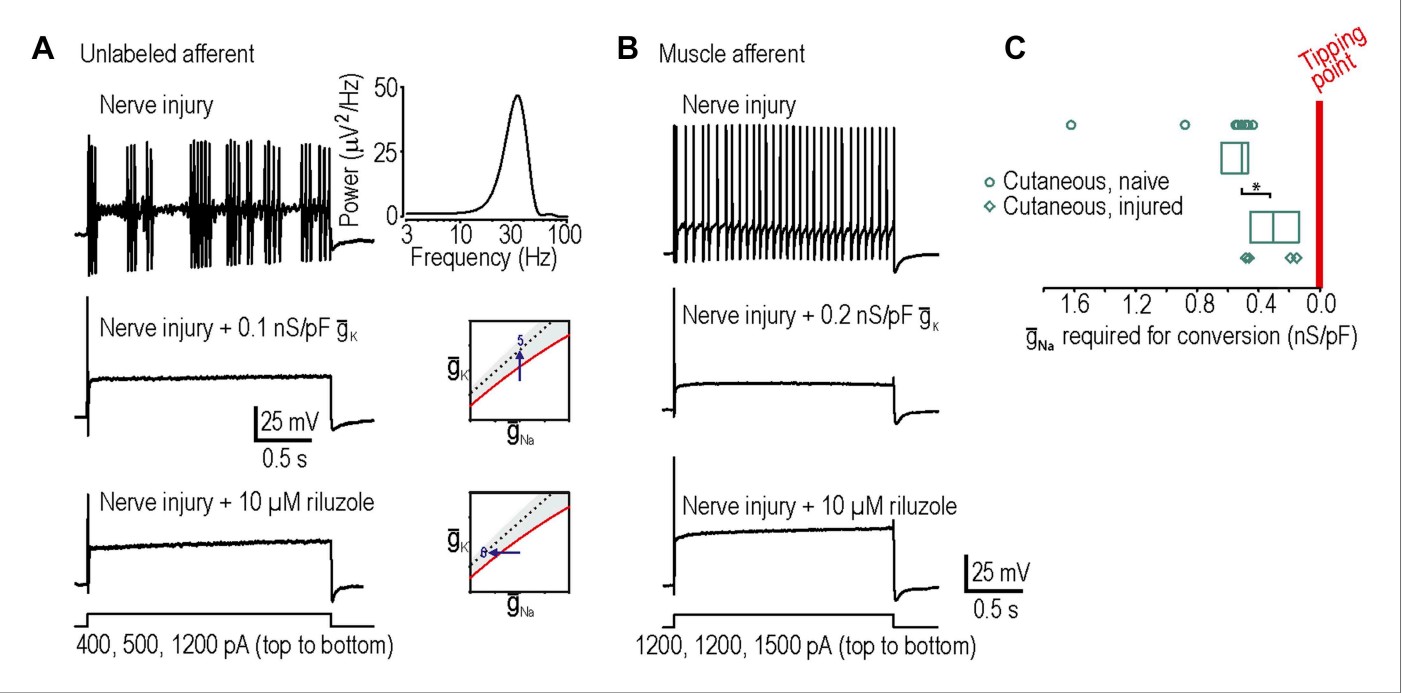

**Figure 4**. Reversal of neuropathic excitability in neurons from nerve-inured rats. (**A**) Typical response of an injured unlabeled primary afferent (top). Repetitive spiking and MPOs were abolished by insertion of a potassium conductance (middle) or by blockade of sodium channels using 10 µM riluzole (bottom), thus confirming predictions 5 and 6, respectively. (**B**) Typical response of an injured muscle afferent whose neuropathic excitability was similarly reversed by increased potassium conductance and reduced sodium conductance. (**C**) Although only 1 of 9 injured cutaneous afferents exhibited repetitive spiking and MPOs, those afferents were nonetheless closer to criticality insofar as they required significantly less virtual sodium conductance than uninjured cutaneous afferents to reach criticality (*p<0.05; Mann–Whitney U test). Points represent data from individual neurons and boxes represent median and 25–75 percentile range.

closer to their tipping point after nerve injury than under control conditions. In the four injured cutaneous afferents to which we added virtual sodium conductance, only 0.30 (range 0.14–0.44) nS/pF of sodium conductance was required to reproduce neuropathic excitability, which is significantly less than the 0.51 (range 0.47–0.63) nS/pF required to reproduce neuropathic excitability in uninjured cutaneous afferents (p<0.05; Mann–Whitney test) (**Figure 4C**). Hence, nerve injury leads to qualitatively altered excitability in only a subset of cells (i.e., ones that cross their tipping point) but the remaining cells are nonetheless quantitatively more excitable (i.e., closer to their tipping point).

## Differential susceptibility of afferent subtypes to developing neuropathic excitability

Using neurons from naive rats, we subsequently verified that muscle afferents, like cutaneous afferents described in **Figure 2**, normally exhibit onset-only spiking (*n* = 27 muscle afferents). Neuropathic excitability was reproduced in 16 of 20 muscle afferents by insertion of virtual sodium conductance (**Figure 5A**), which is a significantly greater conversion rate than 9 of 19 cutaneous afferents (p<0.05; Fisher's exact test). Moreover, we found that the median sodium conductance required to reproduce neuropathic excitability was 0.31 (range 0.25–0.41) nS/pF in muscle afferents compared with 0.51 (range 0.47–0.63) nS/pF in cutaneous afferents, indicating that naive muscle afferents are significantly closer to their tipping point than naive cutaneous afferents (p<0.001; Mann–Whitney test) (**Figure 5B**). Because we cannot isolate the density of native Na$_V$1.3 cannels, we cannot gauge the relative change that this absolute virtual conductance represents. Furthermore, our measurement represents the distance to tipping point along only one dimension; if criticality exists along a boundary in a high-dimensional space (**Figure 1**), then the distance to tipping point may differ significantly along different dimensions. But notably, input resistance did not differ significantly between cutaneous and muscle afferents (p=0.2; unpaired *t* test; 282 ± 47 MΩ in cutaneous afferents vs 377 ± 57 MΩ in muscle afferents). Overall, these data suggest that muscle

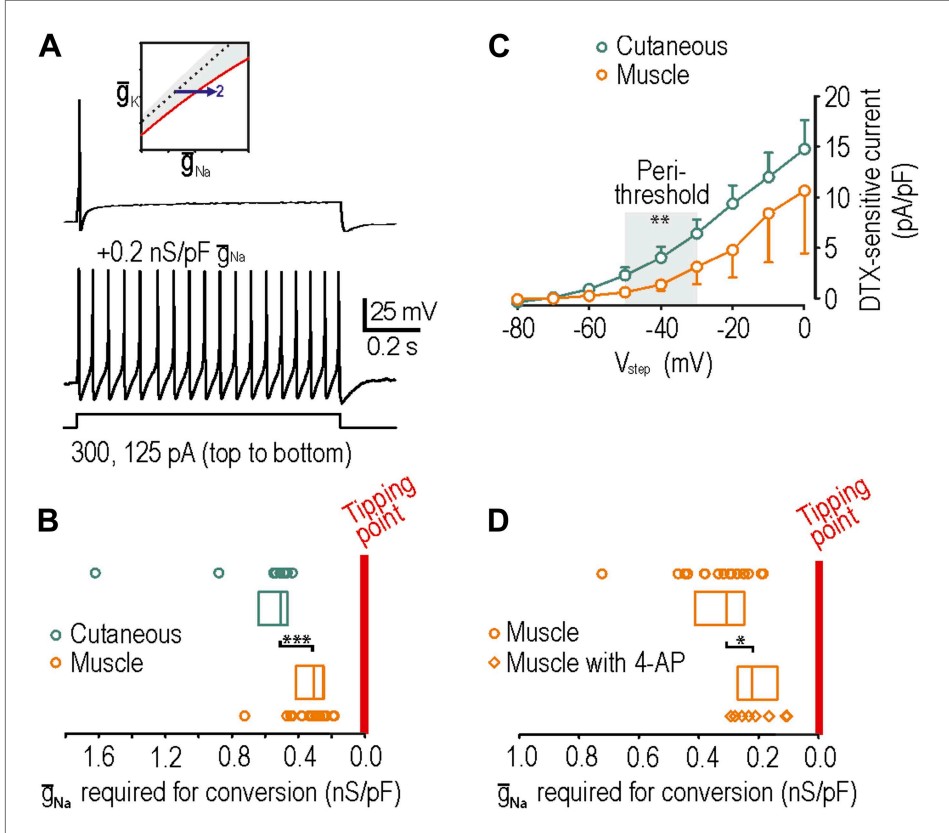

**Figure 5**. Reproduction of neuropathic excitability in naive muscle afferents. (**A**) As predicted, repetitive spiking developed as virtual sodium conductance was increased. (**B**) Uninjured muscle afferents operate significantly closer to criticality than uninjured cutaneous afferents based on the minimum virtual sodium conductance needed to produce repetitive spiking (***p<0.001; Mann–Whitney U test). In **B** and **D**, points represent data from individual neurons and boxes represent median and 25–75 percentile range. These data suggest that muscle afferents are more prone to develop neuropathic excitability after nerve injury because they start off closer to their tipping point, and not necessarily because injury induces a larger change in membrane conductances. (**C**) Based on the observation that 4-AP and dendrotoxin (DTX) produced repetitive spiking in only 1 of 12 muscle afferents, we measured the DTX-sensitive current. The persistent outward current activated at perithreshold voltages (−50 to −30 mV) and blocked by 10 nM DTX was significantly smaller in muscle afferents than in cutaneous afferents (**p<0.01; two-way ANOVA and post-hoc Student-Newman-Keuls test). (**D**) Although 4-AP produced repetitive spiking in only 1 of 12 muscle afferents, it nonetheless moved neurons closer to criticality as evidenced by the minimal virtual sodium conductance needed to cause hyperexcitability after 4AP application (*p<0.05; Mann–Whitney U test).

afferents are more prone to developing neuropathic excitability after nerve injury because they operate closer to their tipping point than do cutaneous afferents.

We also tested whether neuropathic excitability could be reproduced in muscle afferents by reduction of potassium conductance by application or 4-AP or α-dendrotoxin. Contrary to the expectations, 4-AP and dendrotoxin reproduced neuropathic excitability in only 1 of 12 muscle afferents, which is a significantly lower conversion rate than 8 of 15 cutaneous afferents (p<0.05; Fisher's exact test). We hypothesized that this was either (1) because muscle afferents are further than cutaneous afferents from their tipping point or (2) because $K_V1$ expression is lower in muscle afferents. Because hypothesis 1 is inconsistent with data in *Figure 5B*, we tested hypothesis 2 by comparing the density of dendrotoxin-sensitive current in muscle and cutaneous afferents measured in voltage clamp after blockade of sodium channels using 1 µM tetrodotoxin; dendrotoxin was used for these experiments because it is a more selective $K_V1$ blocker than 4-AP. As predicted, the sustained outward current activated at perithreshold voltages (−50 mV to −30 mV) and blocked by 10 nM dendrotoxin was significantly smaller in muscle afferents than in cutaneous afferents (p<0.01; two-way ANOVA and post-hoc

Student-Newman-Keuls test; 1.7 ± 0.6 pA/pF in muscle afferents vs 4.2 ± 0.6 pA/pF in cutaneous afferents after removing variance attributable to voltage) (*Figure 5C*). With fewer $K_V1$ channels available to be blocked, 4-AP and dendrotoxin naturally have a smaller impact on neuronal excitability. However, amongst muscle afferents in which neuropathic excitability was not reproduced by application of 4-AP and in which virtual sodium conductance was subsequently inserted, significantly less sodium conductance was needed to reproduce neuropathic excitability with 4-AP (0.22, range 0.14–0.27 nS/pF) than without 4-AP (0.31, range 0.25–0.41 nS/pF) (p<0.05; Mann–Whitney U test) (*Figure 5D*). This indicates that 4-AP moves muscle afferents toward their tipping point but simply not far enough to reach it, reminiscent of the effects of nerve injury on cutaneous afferents (*Figure 4C*). Moreover, although many factors contribute to regulating excitability ('Introduction'), the relatively low expression of $K_v1$ channels in muscle afferents likely contributes to them existing closer to their tipping point, as shown in *Figure 5B*.

## Discussion

Using computer simulations and experiments, we have demonstrated that primary afferent hyperexcitability associated with neuropathic pain arises through a switch in spike initiation dynamics that occurs when neurons cross a tipping point. Furthermore, there are many different ways to cross that tipping point. The first observation demonstrates criticality, whereas the latter demonstrates degeneracy. Criticality and degeneracy are both important concepts for understanding how complex systems normally operate and how they fail in disease states, but neither concept has been previously applied to help decipher the pathogenesis of neuropathic pain. We will discuss each concept in turn, and then address their relevance for understanding neuropathic pain.

Nonlinear dynamical analysis of our mathematical model predicted that qualitative neuropathic changes in primary afferent excitability—repetitive spiking, MPOs, and bursting—co-develop when subthreshold inward and outward currents become sufficiently unbalanced. According to our theory, the tipping point represents a qualitative change in spike initiation dynamics. By pharmacologically decreasing and/or electrophysiologically increasing candidate conductances, we were able to force primary afferent neurons in different directions across their tipping point, thereby reproducing or reversing neuropathic changes in excitability. These data clearly demonstrate criticality and support our attribution of it to a switch in spike initiation dynamics.

Our theory further predicted, and our experimental data confirmed, that there are multiple ways in which spike initiation dynamics can be altered. To be clear, the tipping point constitutes a unique switch in spike initiation dynamics but there are many different ways to reach that tipping point. Multiple bases for the same outcome constitute degeneracy (*Edelman and Gally, 2001*; *Marder and Taylor, 2011*). Degeneracy prevents emergent network and cellular behaviors from being ascribed to unique ion channel combinations, but therein allows for a disturbance in one type of ion channel to be offset by compensatory changes in other ion channels so that network and cellular function can be robustly maintained (*Flake et al., 2004*; *Howard et al., 2007*; *Marder, 2011*). We focused on manipulating two factors, maximal conductance of subthreshold sodium and potassium conductances, while maintaining all other factors unchanged. Our data clearly demonstrate the degeneracy of neuropathic excitability insofar as distinct manipulations produced equivalent outcomes.

In contrast to our simplified testing paradigm, nerve injury induces numerous molecular changes within primary afferents as evidenced by gene expression profiling (*Costigan et al., 2002*; *Xiao et al., 2002*; *Valder et al., 2003*; *Hammer et al., 2010*; *LaCroix-Fralish et al., 2011*). Some of those changes increase excitability, others are compensatory, and most are irrelevant for (although nonetheless correlated with) hyperexcitability and instead bear on other aspects of cellular function. That said, alterations in cellular excitability cannot be definitively explained without accounting for all 'relevant' changes. This is an insurmountable task if one considers that each neuron experiences different molecular changes and that those changes occur on different molecular backgrounds. However, one can ascertain how close or far a neuron sits from its tipping point based on whether controlled manipulations succeed in forcing the neuron across that point. This requires a theoretical understanding of tipping points (i.e., their nonlinear dynamical basis) to identify telltale signs by which they can be inferred experimentally. It also requires a testing paradigm in which controlled manipulations can be applied, and where unknown changes (differences) do not occur (exist). It is for these reasons that we compared excitability within the same neuron before and after artificial manipulations rather than comparing excitability before and after nerve injury; notably, the latter approach

would have necessitated comparison of excitability in separate neurons, each of which is distinct, and is plagued by countless injury-induced changes, not all of which can be simultaneously measured. Moreover, the abruptness of our manipulations precludes compensatory changes from developing. Thus, our approach represents an innovative alternative to experiments using nerve injury models or molecular–genetic techniques in which channel expression is more slowly up- or down-regulated.

Our results do not reveal a cure for neuropathic pain but, instead, suggest that a paradigm shift is needed in how we approach this task. Criticality and degeneracy may explain features of neuropathic pain that have hitherto eluded explanation. For instance, neuropathic symptoms often develop at long and highly variable delays after the inciting injury or disease. Criticality can explain this insofar as underlying molecular changes can develop occultly, without obvious outward manifestations, until a tipping point is reached, whereupon symptoms develop abruptly because of gross changes in excitability. Furthermore, once near the tipping point, symptoms could wax or wane because of the system moving forward and backward across its tipping point due to subtle fluctuations in underlying factors. From a therapeutic perspective, an obvious goal is to move the system from the 'abnormal' side of the tipping point back to the 'normal' side. Because of degeneracy, there are many ways to cross the tipping point in the forward direction, and an equally large number of ways to cross back in the reverse direction. This last observation is promising, as is the observation that the neuropathic state seems to lie close to the tipping point. But why then is neuropathic pain so notoriously difficult to treat?

The answer to the last question hinges on the answer to another question: Why does neuropathic excitability develop in the first place? Degeneracy ought to convey robustness to the regulation of neuronal excitability since effects of pathological molecular changes can be mitigated by compensatory changes in any one of a multitude of other ion channels. But gross changes in excitability do occur, at least in some neurons, and although those changes were easily reversed by our acute manipulations, prolonged therapies in patients and animal models often have only transient effects. One possible explanation is that the homeostatic regulation of excitability is itself deranged, in effect maintaining the system at an incorrect 'set point' and hijacking degeneracy to paradoxically maintain hyperexcitability. In that scenario, therapeutically manipulating only one misregulated conductance will fail to sustainably reverse hyperexcitability if a deranged homeostatic control mechanism can fall back on other conductances to achieve its misguided goal. Instead, all misregulated conductances need to be coordinately targeted. If epilepsy has a similarly degenerate molecular basis (*Klassen et al., 2011*), it is no wonder that multi-target drugs tend to work well or that polypharmacy with single-target drugs is beneficial (*Kwan et al., 2001*); indeed, combination pharmacotherapy also offers benefits in the treatment of neuropathic pain (*Chaparro et al., 2012*). But, alternatively, the deranged control mechanism itself could be commandeered or its set point adjusted. The idea that neuropathic pain reflects the maladaptive homeostatic response to injury rather than being a direct consequence of injury has started to gain traction (*Costigan et al., 2009*). Our data emphasize the importance of considering the context in which that plasticity occurs, namely, that it occurs within a system that is both complex and degenerate.

To conclude, we reproduced and reversed neuropathic excitability in primary afferent neurons using manipulations designed to force the system across the tipping point that separates normal and neuropathic excitability. Existence of that tipping point demonstrates 'criticality' and speaks to the importance of the nonlinear interactions that give rise to system complexity. Observation that multiple different manipulations can force the system across that tipping point demonstrates 'degeneracy'. Criticality and degeneracy are fundamentally important concepts for understanding how complex systems fail, and how to therapeutically intervene to prevent or reverse those failures. Our theory suggests that treatment strategies for neuropathic pain must move away from single-target-drugs and towards a systems-based approach, capitalizing on degeneracy rather than being thwarted by it.

## Materials and methods

### Simulations

Starting with a two-dimensional Morris–Lecar (ML) model, which is sufficient to produce spikes, we added Hodgkin–Huxley (HH) style conductances for the express purpose of modulating spike initiation dynamics in the ML model. For ML equations, see *Rho and Prescott (2012)*. Parameter values were as follows: $C$ = 2 μF/cm$^2$, $E_{Na}$ = 50 mV, $E_K$ = −100 mV, $E_{leak}$ = −70 mV, $\varphi_w$ = 0.15, $\bar{g}_{fast}$ = 20 mS/cm$^2$, $\bar{g}_{slow}$ = 20 mS/cm$^2$, $g_{leak}$ = 2 mS/cm$^2$, $\beta_m$ = −1.2 mV, $\gamma_m$ = 14 mV, $\beta_w$ = −10 mV, $\gamma_w$ = 10 mV, and $I_{stim}$ was varied. HH conductances were modeled according to

$$I_{Na,K} = \bar{g}_{Na,K} m (V - E_{Na,K}), \tag{1}$$

$$\frac{dm}{dt} = \alpha(1-m) - \beta m, \tag{2}$$

$$\alpha = k_\alpha \frac{\left(\frac{V - V_\alpha}{s_\alpha}\right)}{e^{\left(\frac{V-V_\alpha}{s_\alpha}\right)} - 1}, \tag{3}$$

$$\beta = k_\beta e^{\left(\frac{V-V_\beta}{s_\beta}\right)}, \tag{4}$$

where, $V_{\alpha,\beta} = -24$ mV, $s_{\alpha,\beta} = -17$ mV, and $k_{\alpha,\beta}$ was between 1 and 1.5 ms$^{-1}$. These parameters were chosen to give activation in the perithreshold voltage range (based on the threshold determined by ML parameters). In the interest of simplicity, the activation variable $m$ was not raised to any exponent and nor was inactivation included. The Na$^+$ and K$^+$ currents differed only their reversal potential, where $E_{Na} = +50$ mV and $E_K = -100$ mV. Maximal conductances $\bar{g}_{Na}$ and/or $\bar{g}_K$ were varied in order to modulate spike initiation dynamics and thereby test our hypotheses. Where indicated, we also included a virtual slow AHP current modeled according to

$$I_{AHP} = \bar{g}_{AHP} z (V - E_K), \tag{5}$$

$$\frac{dz}{dt} = \left(\frac{1}{1 + e^{\frac{\beta_z - V}{\gamma_z}}} - z\right) / \tau_z, \tag{6}$$

where. $\beta_z = 0$ mV, $\gamma_z = 5$ mV, $\tau_z = 300$ ms. Setting $\beta_z$ to a suprathreshold voltage ensures a spike-dependent form of adaptation (*Prescott and Sejnowski, 2008*). Gaussian noise with standard deviation 0.3 µA/cm$^2$ and 0 mean was also added, where indicated. Simulations were conducted in XPP and bifurcation analysis was conducted with AUTO using the XPP interface (*Ermentrout, 2002*).

## Animals

All experiments were carried out on adult (200–340 g) male Sprague–Dawley rats (Harlan, Indianapolis, IN or Charles River, Montreal) and were approved by the University of Pittsburgh Institutional Animal Care and Use Committee (protocol number 1108600) and by The Hospital for Sick Children Animal Care Committee (protocol number 22919).

## Neuron labeling

To retrogradely label cutaneous or muscle afferent cell bodies in the dorsal root ganglion (DRG), animals were anesthetized with isoflurane (4% for induction; 2.5% for maintenance) and DiI (Invitrogen, Carlsbad, CA) dissolved in DMSO (170 mg/ml; Sigma–Aldrich, St. Louis, MO) and diluted 1:10 in 0.9% sterile saline was injected into either the hindpaw skin or gastrocnemius muscle. Specifically, to label cutaneous afferents, 10 µl of DiI solution was injected intradermally in 5 sites (2 µl/site) over an area of ~5 mm$^2$ into the glabrous skin of the left hindpaw. To label muscle afferents, 10 µl of DiI solution was slowly injected into five sites of the gastrocnemius muscle of the left leg (2 µl/site); to exclude spurious labeling of cutaneous afferents along the needle track, 10 µl of Fast Blue (1% in sterile saline) was injected intradermally around the intramuscular injection site. Only cells labeled with DiI and not Fast Blue were considered muscle afferents. Neurons were harvested 10–20 days after injections.

## Nerve injury model

A subset of animals received spinal nerve ligation (SNL) (*Kim and Chung, 1992*) 2–5 days before terminal experiments. Under isoflurane anesthesia, paraspinal muscles of the lower lumbar and sacral level were separated to access the area around the L6 process. The L6 process was carefully removed

and the L5 spinal nerve was tightly ligated with 6-0 silk suture. All nerve-injured animals maintained motor function but developed neuropathic pain as inferred by guarding of the affected paw.

## Cell dissociation

To collect DRG neurons, rats were deeply anesthetized by subcutaneous injection of anesthetic cocktail (1 ml/kg of 55 mg/ml ketamine, 5.5 mg/ml xylazine, and 1.1 mg/ml acepromazine) or by isoflurane inhalation (4% for induction; 2.5% for maintenance). DRG were surgically removed (L4 and L5 in naive animals; L5 in nerve-injured animals), enzymatically treated, mechanically dissociated, plated on glass coverslips previously coated by a solution of 0.1 mg/ml poly-D-lysine, and incubated in MEM-BS at 37°C, 3% $CO_2$, and 90% humidity for 2 hr. After incubation, coverslips were transferred to a HEPES-buffered L-15 media containing 10% BS and 5 mM glucose and stored at room temperature. Neurons were studied 2–28 hr after harvesting.

## Electrophysiology

Coverslips with cultured cells were transferred to a recording chamber constantly perfused with room temperature, oxygenated (95% $O_2$ and 5% $CO_2$) artificial cerebral spinal fluid containing (in mM) 126 NaCl, 2.5 KCl, 2 $CaCl_2$, 2 $MgCl_2$, 10 D-glucose, 26 $NaHCO_3$, 1.25 $NaH_2PO_4$. Using gradient contrast optics, neurons with somatic diameter ≥30 µm were targeted for patching as these give rise to myelinated fibers (*Harper and Lawson, 1985*). Cutaneous or muscle afferents were targeted based on epifluorescent illumination of DiI labeling (and exclusion of Fast Blue labeling in the case of muscle afferents). Cells were recorded in the whole-cell configuration with >70% series resistance compensation using an Axopatch 200B amplifier (Molecular Devices; Palo Alto, CA). Electrodes (2–5 MΩ) were filled with a recording solution containing (in mM) 125 $KMeSO_4$, 5 KCl, 10 HEPES, 2 $MgCl_2$, 4 ATP, 0.4 GTP, as well as 0.1% Lucifer Yellow; pH was adjusted to 7.2 with KOH and osmolality was between 270 and 290 mosmol/l. The pipette shank was painted with Sylgard to reduce pipette capacitance. Responses were low-pass filtered at 2 KHz, digitized at 20 KHz using a CED 1401 computer interface (Cambridge Electronic Design, Cambridge, UK), and analyzed offline. Membrane potential (after correction for the liquid junction potential of −9 mV) was adjusted to −65 mV through tonic current injection.

To block native conductances, various drugs were bath applied as reported in the 'Results'. To add virtual conductances, the dynamic clamp technique was applied using Signal 5 software (Cambridge Electronic Design). Conductances were modeled using the same equations as in computer simulations, as described above. To account for differences in cell size, values of inserted conductances were converted to densities by normalizing by membrane capacitance $C$, which was measured from responses to small (≤50 pA) hyperpolarizing current steps, where $C = \tau_m/R_{in}$, $\tau_m$ is the membrane time constant, and $R_{in}$ is input resistance. Notably, given a specific membrane capacitance of 1 µF/cm$^2$, a conductance density of 1 mS/cm$^2$ (as reported for simulations) corresponds to 1 nS/pF (as reported for dynamic clamp experiments).

## Acknowledgements

This work was supported by NIH grant NS074146, a Natural Sciences and Engineering Research Council of Canada Discovery Grant, and scholar awards from the Rita Allen Foundation and the Edward Mallinckrodt Jr Foundation to SAP. SAP is also a Canadian Institutes of Health Research New Investigator. We thank Nicole Scheff and Michael Gold for providing acutely dissociated DRG neurons for pilot studies, Erica Schwartz for instruction on the spinal nerve ligation procedure, Nadine Simons-Weidenmaier for technical assistance, and Tim Bergel for dynamic clamp modifications. We also thank Yves De Koninck, Jerry Gebhart, and Mike Salter for feedback on the manuscript.

## Additional information

### Funding

| Funder | Grant reference number | Author |
|---|---|---|
| National Institutes of Health | R21 NS074146 | Steven A Prescott |
| Natural Sciences and Engineering Research Council of Canada | | Steven A Prescott |

| Funder | Grant reference number | Author |
|---|---|---|
| Rita Allen Foundation | | Steven A Prescott |
| Edward Mallinckrodt Jr Foundation | | Steven A Prescott |
| Canadian Institutes of Health Research | | Steven A Prescott |

The funder had no role in study design, data collection and interpretation, or the decision to submit the work for publication.

## Author contributions

SR, SAP, Conception and design, Acquisition of data, Analysis and interpretation of data, Drafting or revising the article; YZ, Acquisition of data, Analysis and interpretation of data; KYL, Acquisition of data

## Ethics

Animal experimentation: All experiments were carried out on adult (200-340 g) male Sprague–Dawley rats (Harlan, Indianapolis, IN or Charles River, Montreal) and were approved by the University of Pittsburgh Institutional Animal Care and Use Committee (protocol number 1108600) and by The Hospital for Sick Children Animal Care Committee (protocol number 22919).

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
