## [Decision Letter]

[Editors’ note: a previous version of this study was rejected after peer review, but the authors submitted for reconsideration. The two decision letters after peer review are shown below.]

Thank you for choosing to send your work entitled “Injury-induced changes in primary afferent excitability: criticality, degeneracy, and implications for neuropathic pain” for consideration at *eLife*. Your full submission has been evaluated by a Senior editor and 3 peer reviewers, one of whom, Ronald L Calabrese, is a member of our Board of Reviewing Editors, and the decision was reached after discussions between the reviewers. We regret to inform you that your work will not be considered further for publication at this time, although we are willing to consider a thoroughly revised version, which will be treated as an entirely new manuscript. We understand that you may not be interested in this route, and that you may choose to send the present manuscript elsewhere.

There was considerable enthusiasm for the aims of the work but this enthusiasm was dampened by shortfalls in the data. One reviewer wrote “I would like to reiterate that I feel the study has significant merit and if these issues are addressed appropriately, this could represent a major contribution to the field. Nobody else in the field is really doing this type of work, and the conclusions suggest an entirely new way that we should potentially be looking at the mechanisms of neuropathic pain of peripheral origin.“ Resolving the issues however, would require extensive new experiments. Three primary issues were identified:

1) Proper identification of afferents particularly deep tissues afferents. More identified hyper-excitable cutaneous afferents are needed as these are the ones that are thought to be involved in the neuropathic pain rodent model. In general larger n's for all experimental groups with appropriate statistical analyses are needed.

2) Proper controls need to be put in place: e.g., controls for treatments leading to hyper-excitability.

3) For the dynamic-clamp experiments and the scaling of conductance injection, passive properties need to be measured on each single cell submitted to dynamic-clamp. These measurements are needed so that the need conductance for criticality can be scaled for proper interpretation.

The reviews of the outside reviewers are attached in their entirety. If these issues are addressed then a resubmission to *eLife* is encouraged.

*Reviewer*
*1:*

The authors combine theoretical and experimental studies to address the question of criticality and degeneracy in the mechanisms that convert primary afferent neurons from normal single-spike activity to the repetitive firing characteristic of injury-induced neuropathic pain state. The theoretical analyses based on the dynamical systems approach considering just inward and outward spike conductances indicates that multiple mechanisms can move neuron past a critical point that represents the transition from single-spike to repetitive firing. The critical point is in actuality a critical line that itself is subject to movement as other parameters such a leak conductance are varied. Using dissociated dorsal root ganglion (DRG) neurons from rat - some identified as cutaneous afferents - and a combination of pharmacological blockade of K currents and addition of Na currents with dynamic clamp, they demonstrate criticality and degeneracy in neuron neurons in the transition studied. Most exciting is the finding that when using DRG neurons that show injury-induced neuropathic-like repetitive firing from a rat model that they can rescue normal single-spike firing with pharmacological blockade of subthreshold Na current or addition of K current with dynamic clamp.

The writing is succinct and the paper is very easy to follow. The figures are simple but elegant and provide the necessary data in an easily digestible form. The work appears carefully done with sufficient attention to detail and adequate numbers of experiments. The work may prove controversial in the pain community because of its radical approach but the data is just too interesting not to put out there. The controversy will come in the interpretation not in the data itself, and it seems to me that *eLife* should support work that pushes a field in new directions.

Concern:

1) I was intrigued by the notion that some types of afferents live closer to the criticality edge than others. If I could ask for more experiments it would be to look for more identified cutaneous afferents that show repetitive firing in the injury model and to identify the afferent types that do show repetitive firing with tracers if possible.

*Reviewer*
*2:*

In the present article, Ratte and colleagues investigate the criticality and degeneracy of the pathological changes in primary afferent excitability involved in the aetiology of neuropathic pain. Using computer simulation, pharmacological manipulation and dynamic-clamp electrophysiology, the authors show the existence of a tipping point where primary afferent activity qualitatively switches from single-spiking to tonic spiking in response to current injection, therefore demonstrating the existence of criticality in the behavior of these neurons. Then, the authors show that multiple different manipulations can trigger excitability changes similar to those observed in neuropathic pain, therefore demonstrating the degeneracy of the transition between the physiological and the pathological conditions. The authors argue that the existence of multiple equivalent molecular solutions for the transitions between the physiological and pathological states indicates the potential robustness of both of these states and explains, at least in part, the inefficiency of single-target drugs. Therefore, the authors conclude that new drug design strategies, based on the degeneracy principle, may be needed to treat pathologies such as neuropathic pain, in particular the search for drugs that act on multiple-targets.

Overall, this article is very interesting in that it emphasizes the theory that biological systems use very diverse solutions to solve the same problems, which might be one of the main bases of biological robustness. Moreover, the focus of the current article is to investigate not only physiological robustness but also pathological robustness, especially in regard to drug resistance. The experimental design is elegant (although not as innovative as the authors claim), the experiments were suitably performed, and the article is well written. I have one major concern about some of the conclusions drawn in the present study:

The authors claim that the dynamic-clamp experiments performed on cutaneous afferents and deep tissue afferents argue in favor of deep tissue afferents being closer to the tipping point. This conclusion arises from the fact that the injected conductance needed to trigger the pathological behavior of deep tissue afferents is lower than that needed to cross the tipping point in cutaneous afferents. However, this conclusion is biased, especially in the absence of control experiments measuring the intrinsic excitability and conductance levels in native neurons. In other words, one could easily imagine that deeper tissue afferents display higher input resistance values than cutaneous afferents, and that their channel densities (in particular sodium and potassium channels) are lower than for cutaneous afferents. If this is the case, what should be compared is not the absolute conductance value that needs to be injected to trigger or reverse the pathological state, but conductance values that are adjusted for the differences in input resistance and channel densities between the two neuronal types. It is known that input resistance and channel densities are highly variable within a same cell type but also highly variable between cell types. Such differences may explain the results presented in Figure 4 without invoking the differences in distance between the physiological and pathological states in the two cell types analyzed. Although the conclusion drawn by the authors is very interesting and might be true, the absence of control measurements of the native passive and active properties of the two cell types prevents it from being the only interpretation.

*Reviewer*
*3:*

This manuscript by Ratté et al describes theoretical and experimental examination of mechanisms of hyperexcitability of sensory neurons from the rat in the context of nerve injury pain. The work is highly unique to the field, and the findings of the study are quite provocative and have the potential to represent a major advance. There are several issues with the study that reduce the potential impact of the work, however.

1) The entire study is predicated on the reported increase in excitability of sensory neurons from rodents with neuropathic pain. The authors examine ways in which modulation of multiple ion conductances can lead to the generation of this type of hyperexcitability. The problem (and it is a big one) is that the authors' own data fail to demonstrate increased excitability of cutaneous afferents in the context of nerve injury. Thus, only 1/9 cells that innervate the hind paw skin from rats with nerve injury showed repetitive firing - and this is the phenomenon that their theoretical work is meant to be examining. The authors do show that when they examine unlabeled neurons (neurons that do not take up DiI injected into the hind-paw skin), most of these cells (6/9) demonstrate repetitive firing. The major findings were then repeated in these unlabeled neurons, and the findings were consistent. However, the disconnect shines light on the major shortcomings of this study, which are two-fold.

a) The authors do not demonstrate increased incidence of repetitive firing in injured neurons compared to un-injured neurons, and

b) The number of cells in each group is extremely small, given the large degree of heterogeneity among firing properties of rodent sensory neurons. Certainly the number of cells in each group needs to be increased.

2) Many of the comparisons do not have statistical analysis to support the conclusions. Rather, the authors state things like “...was sufficient to produce repetitive spiking and MPOs in 6 of 13 neurons tested with both manipulations. In those 6 neurons, we tested the two manipulations together and found that the combination was sufficient to produce hyperxcitability in 5 neurons.” What does it mean when something happens in 6 of 13 cells? What happened in the other 7 cells? On average, that might suggest that nothing happens. I recognize that there is a high degree of variability, but some statistical analyses are a must here. Or to the point, there are really no “control” experiments – examining cells recorded without the stated manipulation, or using the vehicle in which the drugs were dissolved, for example.

3) The authors record some neurons that were not labeled via DiI injection into the paw. These neurons are called “deep tissue” neurons. This is not correct. The fact that the neurons do not take up dye injected into the paw skin does not mean they are neurons that project to deeper tissues. These should simply be called “unlabeled” neurons. This brings up a very important point. Either the cutaneous afferents are indeed less excitable after injury than deep neurons, or alternatively, perhaps the DiI is causing changes to the firing properties. The appropriate comparison would be to inject deeper tissues with DiI and record from those neurons to make sure that the observed differences are indeed due to target of innervation as opposed to the presence of the dye. The data provided do *not* demonstrate that “deep tissue afferents are more prone to qualitative changes in excitability....” as the authors claim.

[Editors’ note: what now follows is the decision letter after the authors submitted for further consideration.]

Thank you for sending your work entitled “Injury-induced changes in primary afferent excitability: criticality, degeneracy, and implications for neuropathic pain” for consideration at *eLife*. Your article has been favorably evaluated by a Senior editor and 3 reviewers, one of whom, Ronald L Calabrese, is a member of our Board of Reviewing Editors.

The Reviewing editor and the other reviewers discussed their comments before we reached this decision, and the Reviewing editor has assembled the following comments to help you prepare a revised submission.

Small concern:

Please clarify the model a bit more. Is it correct that spiking is generated in the model by the standard M-L equations and that the added HH Na and K currents are strictly subthreshold; i.e., they do not produce spikes? This should be stated explicitly in both Results (at the very beginning of Results) and in Methods. Also lower panel of Figure 1 is very confusing as it is not mapped onto the model or mentioned in the text.

---

## [Author Response]

[Editors’ note: the author responses to the first round of peer review follow.]

*1A) Proper identification of afferents particularly deep tissues afferents*.

We have added new experiments from muscle afferents labeled in naïve and nerve-injured animals. These new data appear in Figures 4 and 5.

*1B) More identified hyper-excitable cutaneous afferents are needed as these are the ones that are thought to be involved in the neuropathic pain rodent model*.

We started under the assumption (and conveyed the impression in the original text) that cutaneous afferents are primarily responsible for allodynia in neuropathic pain conditions, which is why we deliberately targeted them with DiI labeling in our initial experiments. In fact, muscle afferents are more prone to developing hyperexcitability (e.g. Michaelis et al. J Neurosci 2000), which is confirmed by our new data. [41] reported repetitive spiking in only 6 of 33 injured cutaneous neurons, which is comparable to the proportion we observed (1 of 9; p = 1, Fisher’s exact test); by comparison, they observed repetitive spiking in 18 of 21 injured muscle afferents, which is significantly more than the proportion we observed (11 of 20; p < 0.05, Fisher’s exact test) but is still reasonably similar given a variety of methodological differences. In any case, 11 of 20 muscle afferents showing repetitive spiking after nerve injury is a significantly higher proportion than 0 of 27 muscle afferents showing repetitive spiking under control conditions (p < 0.001, Fisher’s exact test). Overall, additional experiments in cutaneous afferents are not warranted given past studies and since our new experiments in muscle afferents now demonstrate that a high proportion of such afferents become grossly hyperexcitable after nerve injury.

Importantly, mechanical hypersensitivity is tested in rodents by poking the foot with von Frey filaments, which also activates underlying muscle afferents (e.g., Jankowski et al. J Neurophysiol 2013); thus, muscle afferent hyperexcitability very likely contributes to mechanical hypersensitivity in animal models. The difference between punctate stimulation and the light brushing used in clinical studies is a source of controversy (e.g., Mogil Nat Rev Neurosci 2009), but that is beyond the scope of our study. It should also be kept in mind that cutaneous afferents, although not exhibiting gross hyperexcitability, are nonetheless quantitatively more excitable after injury (Figure 4) and that muscle afferent hyperexcitability could alter the central processing of cutaneous input.

Based on the newly added data from muscle afferents from both control and nerve-injured animals, and based on extensive changes to the text, we think this concern has been thoroughly addressed.

*1C In general larger n's for all experimental groups with appropriate statistical analyses are needed*.

We have more or less doubled the number of cells now reported in the paper. This is largely due to the new data from muscle afferents, but certain experiments were also repeated in cutaneous afferents. We have also included statistical analysis wherever appropriate.

Although we are not certain it is necessary, we have compared our conversion rates against estimated “chance” conversions rate based on comparing the pattern of excitability at the beginning and end of recording from each cell, without any manipulation. No cell ever spontaneously changed to a neuropathic pattern of excitability over the course of recording. If we insert those 0 conversion rates into a contingency table and compare against the conversion rate after 4-AP application or virtual sodium current insertion, Fisher’s exact tests always shows significance. Thus, we can say that our manipulations are significantly more successful in reproducing neuropathic excitability than expected by chance. We have added this statistical analysis to the revised text, although we worry that it might be seen as rather trivial.

Given technical limitations and various biological factors, single manipulations sometimes failed to convert the excitability (i.e., we did not achieve a 100 % forward conversion rate, although we did achieve a 100% reversal rate). When we failed to produce neuropathic excitability by a single manipulation like 4-AP or nerve injury, we have now quantitatively shown that the first manipulation moved the system closer to its tipping point as assessed by the reduction in the second manipulation needed for the system to finally cross that tipping point (i.e., the minimum necessary sodium conductance). This shifting is now quantified Figures 3, 4 and 5 in the revised manuscript.

All of the important comparisons have now been tested statistically and all results were significant. This leads us to believe that the current sample sizes, many of which have indeed been increased since the original submission, are large enough to support our claims. Moreover, the text and figures have been revised to help clearly convey the statistical results.

*2) Proper controls need to be put in place: e.g., controls for treatments leading to hyper-excitability*.

With respect to our nerve injury model, we chose spinal nerve ligation (SNL) because it has been used for > 20 years and, compared with subtler pain models, is known to consistently cause rapid-onset and robust mechanical hypersensitivity (e.g., Kim et al. Exp Brain Res 1997). Given that the SNL model has been thoroughly validated and that we are using the procedure solely to give us hyperexcitable neurons with which we can test predictions 5 and 6, we consider the sham control to be unnecessary (and therefore unethical since it entails animals experiencing untreated post-operative pain). The fact that we did not observe grossly hyperexcitable cutaneous afferents after nerve injury might suggest that our SNL procedure was ineffective, but this is disproven by three observations: (1) nerve-injured animals exhibited behavioral evidence of pain, (2) the cutaneous afferents were quantitatively hyperexcitable, i.e., closer to their tipping point, and (3) the same procedure caused gross hyperexcitability in a sizeable fraction of muscle afferents. This last point is arguably the most important and is now clearly demonstrated in the revised manuscript. There is, therefore, very clear evidence that the SNL procedure worked.

We assume this point might also relate to the second major concern expressed by reviewer 3. He/she wrote: *What does it mean when something happens in 6 of 13 cells? What happened in the other 7 cells? On average, that might suggest that nothing happens. I recognize that there is a high degree of variability, but some statistical analyses are a must here. Or to the point, there are really no “control” experiments – examining cells recorded without the stated manipulation, or using the vehicle in which the drugs were dissolved, for example*.

Most of our key demonstrations rely on within-cell comparisons that were intended to remove effects of cellular heterogeneity, DiI labeling, etc. We routinely turned dynamic clamp on and off to verify its effects, and we washed out drugs after application to demonstrate reversibility. To be very clear, we always examined the cell without the stated manipulation for comparison with the manipulation. As for vehicle controls, the only drug not soluble in water is riluzole; the small amount of DMSO used to dissolve the riluzole has no measurable effect on neuronal excitability.

As to what happens to cells not converted by 4-AP, we now show in both cutaneous afferents (Figure 3) and muscle afferents (Figure 5) that 4-AP, if it did not cause repetitive spiking, has nonetheless moved the neuron closer to its tipping point insofar as significantly less virtual sodium current needs to be added to cause conversion, reminiscent of the effects of nerve injury on cutaneous afferents (Figure 4). We cannot do the converse experiment since it is impossible to precisely quantify the amount of potassium current blocked in current clamp experiments. Nonetheless, based on an unsuspected difficulty with pharmacologically converting muscle afferents (despite the ease of converting them with virtual sodium current), we conducted voltage clamp experiments to test whether muscle afferents express less dendrotoxin-sensitive potassium current. We found that this was indeed the case (Figure 5) With respect to whether DiI or its vehicle may account for differences between labeled and unlabeled neurons, Zhang et al. (J Neurosci Methods 2007) showed that retrograde labeling with up to 5 % DiI (we used 1.7 %) does not affect passive or active membrane properties. In any case, this becomes a moot point since we now compare labeled cutaneous afferents with muscle afferents that are also labeled with DiI.

*3) For the dynamic-clamp experiments and the scaling of conductance injection, passive properties need to be measured on each single cell submitted to dynamic-clamp. These measurements are needed so that the need conductance for criticality can be scaled for proper interpretation*.

We completely agree and have done this in the revised manuscript. We calculated the membrane capacitance from the measured input resistance and membrane time constant. All dynamic clamp conductance values have been normalized by membrane capacitance. The same normalization was applied to new voltage clamp data.

Reviewer 1:

*1) I was intrigued by the notion that some types of afferents live closer to the criticality edge than others. If I could ask for more experiments it would be to look for more identified cutaneous afferents that show repetitive firing in the injury model and to identify the afferent types that do show repetitive firing with tracers if possible*.

We have added extensive new data on labeled muscle afferents. Consistent with past studies, we have confirmed that a significant proportion of muscle afferents are rendered grossly hyperexcitable by nerve injury. This is entirely consistent with our new data showing that uninjured muscle afferents are significantly closer to their tipping point than cutaneous neurons, based on the minimal amount of virtual sodium current needed to produce repetitive spiking. However, we also found that 4-AP application was significantly less likely to produce repetitive spiking in muscle afferents than in cutaneous afferents. This prompted additional experiments which led to a novel finding that muscle afferents have a lower density of dendrotoxin-sensitive potassium current, which may explain why these neurons operate closer to their tipping point. Although these new findings are interesting, especially for those in the pain field, we have tried to retain a clear focus on our initial hypotheses concerning degeneracy and criticality. We hope to have struck a good balance in this revised paper.

Reviewer 2:

*1) The authors claim that the dynamic-clamp experiments performed on cutaneous afferents and deep tissue afferents argue in favor of deep tissue afferents being closer to the tipping point. This conclusion arises from the fact that the injected conductance needed to trigger the pathological behavior of deep tissue afferents is lower than that needed to cross the tipping point in cutaneous afferents. However, this conclusion is biased, especially in the absence of control experiments measuring the intrinsic excitability and conductance levels in native neurons. In other words, one could easily imagine that deeper tissue afferents display higher input resistance values than cutaneous afferents, and that their channel densities (in particular sodium and potassium channels) are lower than for cutaneous afferents. If this is the case, what should be compared is not the absolute conductance value that needs to be injected to trigger or reverse the pathological state, but conductance values that are adjusted for the differences in input resistance and channel densities between the two neuronal types. It is known that input resistance and channel densities are highly variable within a same cell type but also highly variable between cell types. Such differences may explain the results presented in*
Figure 4
*without invoking the differences in distance between the physiological and pathological states in the two cell types analyzed. Although the conclusion drawn by the authors is very interesting and might be true, the absence of control measurements of the native passive and active properties of the two cell types prevents it from being the only interpretation*.

Based on new data in labeled muscle afferents, we now compare the input resistances in muscle and cutaneous afferents and show that they do not significantly differ. We have normalized all dynamic clamp conductance values by capacitance, and so now express those conductance values as densities (i.e., nS/pF). Statistically, uninjured muscle afferents require significantly less virtual sodium conductance than cutaneous afferents to be converted (Figure 5). To be clear, these data are based on experiments in naïve animals and show only that muscle afferents are significantly closer to their tipping point based on this one manipulation; indeed, when thinking in terms of a high-dimensional space, a system may be very close to its tipping point in one dimension but very far it in the other dimensions. We think this is a very interesting topic, but dealing with it here will quickly exceed the intended scope of the current paper. These issues are nonetheless mentioned in the revised text.

Adjusting (or expressing) the virtual conductance value based on native conductance densities is an interesting suggestion but it is technically very challenging. We have now conducted voltage clamp experiments to compare the density of dendrotoxin-sensitive potassium current in muscle and cutaneous afferents. We found the current density to be significantly less in muscle afferents, which is consistent with muscle afferents operating closer to their tipping point, but again, does not exclude other differences. In any case, if the sodium channel mimicked by dynamic clamp was upregulated by nerve injury (and all other conductances remained unchanged) then a certain amount of upregulation could cause gross hyperexcitability amongst muscle afferents whilst failing to cause gross hyperexcitability amongst cutaneous afferents. We express “upregulation” of the virtual channel in absolute terms as there is no virtual channels in the control condition, and because we are not strictly mimicking a native Na_v_1.3 channel and nor could we strictly isolate the density of that channel in order to express the upregulation in relative terms.

In summary, because we do not know and cannot measure all of the possibly relevant native conductance densities (let alone conduct the dynamic clamp experiment in the same neuron), we opted for a minimalist strategy. This leads to a novel suggestion that proximity to tipping point may be a contributing factor for why certain cell types develop gross hyperexcitability after nerve injury while other cell types do not. We cannot exclude other pre-existing differences or differential injury-induced (pathological or compensatory) changes. Quantifying the contribution of each factor would be a difficult task and the interpretation of such results would be subject to many caveats. Overall, we want to introduce the distance-to-tipping-point idea, as we think it helps quantify certain demonstrations that are directly relevant to our main theses, but we think it wise to keep those demonstrations as simple as possible.

Reviewer 3:

*1) The entire study is predicated on the reported increase in excitability of sensory neurons from rodents with neuropathic pain. The authors examine ways in which modulation of multiple ion conductances can lead to the generation of this type of hyperexcitability. The problem (and it is a big one) is that the authors' own data fail to demonstrate increased excitability of cutaneous afferents in the context of nerve injury. Thus, only 1/9 cells that innervate the hind paw skin from rats with nerve injury showed repetitive firing – and this is the phenomenon that their theoretical work is meant to be examining. The authors do show that when they examine unlabeled neurons (neurons that do not take up DiI injected into the hind-paw skin), most of these cells (6/9) demonstrate repetitive firing. The major findings were then repeated in these unlabeled neurons, and the findings were consistent. However, the disconnect shines light on the major shortcomings of this study, which are two-fold*.

*a) The authors do not demonstrate increased incidence of repetitive firing in injured neurons compared to un-injured neurons,*
*and*

*b) The number of cells in each group is extremely small, given the large degree of heterogeneity among firing properties of rodent sensory neurons. Certainly the number of cells in each group needs to be increased*.

With respect to point (a), we now show that there is indeed a significant increase in the incidence of repetitive spiking among muscle afferents. This is consistent with past publications (see response to consensus report). But we are unclear if the reviewer is questioning whether gross hyperexcitability in either type of afferent actually causes neuropathic pain. This is obviously important, but we think we are on safe ground. For instance, blocking peripheral input relieves pain for the duration of the block (e.g., Gold and Gebhart, Nat Med 2010), which argues that peripheral input is necessary to cause pain although it may not be sufficient since extensive plasticity also develops centrally. As explained in the text, myelinated afferents are believed to subserve the mechanical hypersensitivity (or allodynia) associated with neuropathic pain. A switch from onset-only spiking to repetitive spiking would dramatically amplify signaling. That said, cutaneous mechanosensors are the most likely to be directly involved, and yet neither we nor others observed gross hyperexcitability amongst those neurons. We have shown in this study that those neurons do have a critical point and that they are moved towards that critical point by nerve injury; it is conceivable that a higher proportion of cutaneous afferents cross that critical point under the conditions that exist in vivo. It is also possible that input from grossly hyperexcitable muscle afferents drives central heterosynaptic plasticity or summates with the input from cutaneous afferents (e.g., Nagi et al. J Physiol 2011) and thus contributes to mechanical allodynia in an indirect way.

The general consensus within the pain research community is that peripheral input is key and, therefore, that primary afferent hyperexcitability is a causal factor in many forms of neuropathic pain. Our study aims to advance understanding of that hyperexcitability, especially regarding criticality and degeneracy. Our new data on muscle afferents will, we hope, dispel the reviewer’s concerns.

With respect to point (b), please see response to point 1C in the consensus report.

*2) Many of the comparisons do not have statistical analysis to support the conclusions. Rather, the authors state things like “...was sufficient to produce repetitive spiking and MPOs in 6 of 13 neurons tested with both manipulations. In those 6 neurons, we tested the two manipulations together and found that the combination was sufficient to produce hyperxcitability in 5 neurons.” What does it mean when something happens in 6 of 13 cells? What happened in the other 7 cells? On average, that might suggest that nothing happens. I recognize that there is a high degree of variability, but some statistical analyses are a must here. Or to the point, there are really no “control” experiments – examining cells recorded without the stated manipulation, or using the vehicle in which the drugs were dissolved, for example*.

Please see responses to points 1C and 2 in the consensus report. We believe that our revisions have thoroughly addressed this concern.

*3) The authors record some neurons that were not labeled via DiI injection into the paw. These neurons are called “deep tissue” neurons. This is not correct. The fact that the neurons do not take up dye injected into the paw skin does not mean they are neurons that project to deeper tissues. These should simply be called “unlabeled” neurons. This brings up a very important point. Either the cutaneous afferents are indeed less excitable after injury than deep neurons, or alternatively, perhaps the DiI is causing changes to the firing properties. The appropriate comparison would be to inject deeper tissues with DiI and record from those neurons to make sure that the observed differences are indeed due to target of innervation as opposed to the presence of the dye. The data provided do* not *demonstrate that “deep tissue afferents are more prone to qualitative changes in excitability....” as the authors claim*.

Although we have left in sample traces from an unlabeled neuron (now using the nomenclature recommended by the reviewer), we have added substantial new data from labeled muscle afferents and now compare identified cutaneous afferents and identified muscle afferents. The muscle afferents were labeled with DiI, and so DiI labeling is not a confounding factor in the comparison. Based on the new analysis, we think we can safely suggest that muscle afferents operate closer to their tipping point, which may contribute to them being more prone to qualitative changes in excitability.

[Editors’ note: the author responses to the re-review follow.]

*Please clarify the model a bit more. Is it correct that spiking is generated in the model by the standard M-L equations and that the added HH Na and K currents are strictly subthreshold; i.e., they do not produce spikes? This should be stated explicitly in both Results (at the very beginning of Results) and in Methods. Also lower panel of*
Figure 1
*is very confusing as it is not mapped onto the model or mentioned in the text*.

We have rewritten the opening paragraph of the Results and relevant parts of the Methods. We are reluctant to refer to HH currents as “strictly” subthreshold since they continue to be active at suprathreshold voltages, but it is true that they are not necessary for spike generation. We hope the revised text is clear in explaining that addition of an inward or outward HH conductance active at voltages near threshold changes how the fast and slow currents interact during the spike initiation process.

With respect to the lower panel in Figure 1, we now explain in the legend that τ = (α+β)^-1^.